# Palliative Care Needs Assessment in the Population Living in Mayotte: SPMAYOTTE, a Qualitative Study Conducted with 62 Patients, Caregivers and Healthcare Professionals

**DOI:** 10.3390/ijerph19063540

**Published:** 2022-03-16

**Authors:** Raphaël Benoist, Philippe Walker, Karine Allain-Baco, Régis Aubry

**Affiliations:** 1Mobile Palliative Care Unit, Centre Hospitalier Universitaire de La Réunion, Site Félix Guyon, 97400 Saint Denis, France; 2Mobile Palliative Care Unit, Centre Hospitalier Ouest Réunion, 97460 Saint Paul, France; walker.p@ch-gmartin.fr; 3Mobile Palliative Care Unit, Centre Hospitalier de Mayotte, 97600 Mamoudzou, France; k.allain@chmayotte.fr; 4INSERM CIC 1431, Centre Hospitalier Universitaire Jean Minjoz, 25000 Besançon, France; raubry@chu-besancon.fr

**Keywords:** palliative care, patient preference, caregivers, needs assessment, health personnel, Indian Ocean Islands, Comoros, cultural competency

## Abstract

Background/Aim: Mayotte is a French island in the Indian Ocean. There is no palliative care structure in this territory. The island and its population have specific characteristics: insularity, poverty, coexistence of modern and traditional medicine, importance of religion (Islam) and the presence of many foreigners without health insurance. The aim of this study is to determine the palliative care needs of the Mayotte population and propose the establishment of an appropriate service. Methods: A qualitative study was conducted in Mayotte using interviews with patients and their caregivers, and focus groups were conducted with healthcare professionals involved in their care. Patients requiring palliative care were identified and recruited from the hospital or the patient’s home by healthcare professionals. Results: A total of 62 people participated in the study between May and June 2019. The needs expressed were analysed and then grouped into categories: access to medical care (especially at home), management of physical symptoms (analgesia) and psychological symptoms, organisation of care (coordination between healthcare professionals) and training of healthcare professionals (pain management, palliative care, interculturality and translation), taking into account cultural and religious aspects. Regarding the foreign population, the categories were: improving access to healthcare, access to the social protection system and daily living conditions (transport, food and accommodation). Conclusions: The specific needs of the population, assessed through the study, have led to an original proposal, which differs from the usual structures of palliative care in France.

## 1. Introduction

Mayotte (approximately 250,000 inhabitants) has been a French overseas department since 2011 [1] and is lagging behind in many areas, including legal, health and economic. The island is faced with strong migratory pressure and significant demographic growth (+3.8% per year) [1,2]: 55% of the population is under 20 years old, 40% of the population is composed of foreigners and 95% of whom are Comorians, often residing illegally [3]. The consequence of this demographic situation is an increase in community tensions and insecurity, felt by both the local (French) population and the immigrants. The latter have a permanent feeling of insecurity linked to the lack of social protection, an obstacle to access to healthcare and the high risk of expulsion from the territory [4].

### 1.1. Cultural Aspects

Islam is practised by 95% of the Mayotte’s population [5]. Generally speaking, Mahoran society favours the community over the individual, solidarity over autonomy and spiritual values over material rules [6]. Thus, even if illness remains a private event, it is also a social reality [7]. Moreover, there is no clear opposition between the physical and the metaphysical, the profane and the sacred. Thus, the therapeutic quest does not stop at the correction of physical disorders, it aims at restoring a balance, at reassuring the patient in his individual and collective identity: social, religious, physical and psychological [8]. Depending on the origin of the illness, whether it comes from the “social, material or supernatural world” [6], the population has recourse to different medicines. Traditional medicine and spiritual references permeate the population’s approach to illnesses, especially serious illnesses and the end of life [5].

### 1.2. Health Aspects

Mayotte’s precarious situation has led to the need to “catch up” in terms of healthcare [5,9]. Chronic diseases are more frequent than in mainland France, as is overall perinatal and maternal mortality [1]. In addition, life expectancy is about 8 years lower for women and 5 years lower for men [1]. The infrastructures in place rely almost exclusively on the public finances of the French state. They are primarily designed to deal with emergencies [10]. The available healthcare is based on a network of 14 public dispensaries, 4 inter-communal hospitals, 21 maternal and child protection centres and the Mayotte Hospital in Mamoudzou (Figure 1). The primary care sector is very poorly developed, with, for example, 10 times fewer private general practitioners (GPs) than in mainland France [11].

### 1.3. A Serious Delay in Palliative Care

Despite the legal obligation to provide access to palliative care in France since 2019 [12], Mayotte remains the only French department without a palliative care structure. The World Health Organization (WHO) notes this lack by ranking the island, although French, at the lowest level of access to palliative care alongside developing African and Pacific Island countries [13]. While the French administration wishes to develop a palliative care project in Mayotte, it is the specific characteristics (social, anthropological, cultural, demographic and geographical) of this island that led us to conduct a study among the population living there. French regulations governing the organisation of palliative care [14] suggest an assessment of local needs in order to organise the healthcare on offer. Thus, rather than considering “copying” a model straight from France, we wanted to know more precisely the needs of this population in order to adapt the care offer.

The main objective of the SPMAYOTTE study was to determine the needs of the population living in Mayotte in the area of palliative and end-of-life care. The secondary objective was to propose ways to enable institutions to develop a relevant palliative care offer.

## 2. Materials and Methods

### 2.1. Design

This was a descriptive and comprehensive qualitative study, based on semi-structured interviews. In order to obtain a wide range of views and experiences, it combined three lines of investigation: a survey with patients, another survey with family caregivers and finally a survey with healthcare professionals. The general principle of the surveys was to assess the perceived needs for palliative care in relation to patient care.

The methodological framework of this public health approach to health needs assessment was already published in 2021 [15].

### 2.2. Setting and Selection of Participants

This study was conducted in Mayotte by two male researchers, doctors specialised in palliative care, who did not know each other beforehand. They were practising in another French overseas department (Reunion Island) and were trained in the technique of semi-structured interviews. A local referring practitioner (named “referrer” in the article) was involved, responsible for organisational logistics, establishing links with patients and their family caregivers, with healthcare providers and with local administrations. Two interpreters were available throughout the study to translate the interviews.

The inclusion criteria were as follows:-Patient survey: To be designated by one of their healthcare practitioners as being in a palliative situation; have a level of understanding deemed sufficient to participate in the study and be able to give consent; no age criteria or specific pathology.-Family survey: To be nominated by a recruited patient and identified by the patient as a resource person; have a level of understanding deemed sufficient to participate in the study and be able to give consent.-Healthcare professionals’ survey: Should have usually managed a patient included in the study or have expressed a wish to participate in the study as they are regularly confronted with this type of situation (without necessarily having a patient included at the time of the study); have given their consent to participate in the research.

### 2.3. Recruitment

Potential participating patients were identified by their healthcare providers after receiving a letter from the Mayotte Regional Health Agency (ARS). This letter was sent to all health professionals on the island. It informed them of the study and asked them to report, by telephone or email, one or more palliative situations of patients in their care. To help professionals identify a palliative situation, the simple “surprise question” tool was used (would you be surprised if the patient died within a year?”) [16,17]. Among the patients identified, the sampling paid attention to the diversification of criteria, such as nationality (Mahoran or Comorian), main pathology, access to social welfare and place of residence. The population of healthcare professionals should also be diverse (geographical origin, profession and type of practice) [18]. All participants were contacted by telephone once to be informed of the existence and objectives of the study and to obtain their oral consent to participate. A second recorded oral consent was given before the interviews began.

### 2.4. Data Collection

The interviews were conducted between March and June 2019 with the help of an interview guide specific to each population (patients, family and healthcare professionals), compiled from a literature review and discussions with a scientific committee. It included an introductory sentence, a main question (need for palliative care) and secondary probing questions to address the various issues and obtain more complete answers.

The interviews were to be conducted as follows:-Patient survey: individual interviews, in the patient’s usual place of residence;-Family survey: individual or group (focus group) interviews, in the usual place of residence;-Healthcare professionals survey: individual or focus group interviews, in the usual place of work.

When possible, group interviews were chosen because they use group interaction between participants to generate data. In this way, the dynamics of the participants’ interaction could bring out as many themes and concepts related to the research topic as possible. As well, dissension and consensus could emerge in a fairly “natural” way [19,20]. For each survey, the first interview was conducted jointly by the two interviewers. Subsequent interviews were conducted by either one or the other, depending on their availability.

### 2.5. Data Analysis

The interview data were collected by recording (digital dictaphone), after systematic consent from the participants. The full content and verbatim transcripts were transcribed on Word^®^. The number of interviews to be carried out depended, for each of the three surveys, on the saturation of the data (assessed by the two researchers as the interviews were carried out and analysed). The data were coded using the semantic content analysis method with the help of Word^®^ tables. The codes were then grouped into different themes, which in turn were grouped into categories that gave rise to concepts. A theoretical system responding to our problems could emerge while respecting the methodological principles of grounded theory [21,22]. The analysis was carried out initially by one researcher and then a second analysis by the other researcher allowed for a cross-referencing of the data (triangulation).

### 2.6. Ethics

This study was approved by the ethics committee of the university hospital of Besançon on the 3 April 2019. A simplified declaration to the National Commission for Information and Liberties (CNIL) was sent, in accordance with the conformity of this type of protocol (MR-004) [23]. The anonymity of participants was preserved by assigning a code to each participant. The confidentiality of the exchanges was guaranteed, the data collected by the recordings were kept and only accessible by the two researchers. The request for recorded oral consent seemed more legitimate given the cultural aspects and the high level of illiteracy of the population. A time for listening and support as well as psychological follow-up was offered after the interviews were conducted. Participants could withdraw from the study without justification.

## 3. Results

### 3.1. Population

A total of 34 interviews were conducted with 62 people, for 31 recordings, between May and June 2019. They lasted between 10 minutes and 1 hour, the average being 29 minutes. Thirteen patients in palliative care were identified to participate in the study. After verification and exchanges between the healthcare professionals and the referrer, no patients were excluded as all met the criteria defined by the protocol. In practice, the organisation and implementation led to most of the time conducting patient interviews in the presence of their families and vice versa. Descriptive elements of the interviewed population are gathered in Table 1. Details of the coding of the family and patient interviews are given in the Table A1, and those relating to the coding of healthcare professionals are shown in the Table A2. In three situations, two interviews were conducted during the same recording (P8–P10–F6), which explains why the number of recordings is lower (31) than the number of interviews (34). Two family interviews could not be conducted because the relatives were absent or could not be reached (P2–P9). The interviews took place at home for the patients and their families. For the healthcare professionals, they took place in their place of work in hospital (9), in their clinics (4) and in the home of a patient (1). The patients interviewed were almost all suffering from advanced cancers (8/9). Most of the close relatives interviewed had a family link with the patient (e.g., children, spouses, other). The nationality of the relatives was indicated because it could have important consequences on the support and needs expressed, particularly because of access to social welfare.

### 3.2. Results of the Study

The interviews were coded to allow for anonymisation. The results were processed initially by group (patient, family/caregivers and healthcare professionals). The codes used had the following meaning: I = interviewer, P = patient, F = family or caregiver, N = nurse and D = doctor. The summary of the needs expressed is shown in the Table 2.

#### 3.2.1. The Question of Administrative Status and Non-Affiliation

Most of the time, the healthcare journey began on an island of the Comoros. As with the patients, some of their relatives were able to talk about this “health trip”:


*“F: […] we thought it was possible to operate. I: was it the doctor in the Comoros who said it would be better for her to come here? F: yes, he said it would be better to move her elsewhere, to a place where there are many doctors with skills…”*
(Interview F3)

Among the nine patients interviewed, seven were in an illegal situation in Mayotte. When they arrived on Mahoran territory, most often after a crossing in a *kwassa kwassa* (small fishing boats that allow people to cross to Mayotte at night via smugglers), there was still major anxiety related to the fear of expulsion. Then, the difficulties lay in the absence of access to social welfare (as a consequence of not obtaining a residence permit). Access to care for the diagnosis and treatment of their illness was limited by this obstacle. There could be a feeling of inequity regarding access to care and the fear of being badly perceived as a foreigner:


*“F: It’s culturally linked, we know that in Mayotte, when you’re a foreigner, especially when you’re in an illegal situation, it has a strong impact on the relationship that people have with you.”*
(Interview F7)

Moreover, this population had no possibility of home care, in particular, medical follow-up by the community teams (due to their administrative situation). Sometimes the only solution for “quality” care was, paradoxically, to extend the hospital stay (indeed, the public hospital receives any patient requiring hospitalisation, even if they are in an illegal situation). Returning home gave the impression of poor-quality care for the healthcare professionals, with major discomfort at the end of life for this population, which is confronted with financial and material difficulties of all kinds as well as is socially isolating.

In the end, health professionals often expressed the hope that an administrative assistance measure would enable them to have access to appropriate care in acceptable conditions without interruption until the end of their illness.


*“D: But then we have huge needs in community care, enormous. And that’s where we’re going to have to try to make things clear, because there’s a double standard between those who have welfare and those who don’t. So, if they have, well, there will be returns home that are not perfect but that we can manage. But if they don’t have welfare, it’s… we let them get into really… disastrous situations… and it’s catastrophic…”*
(interview S3)

#### 3.2.2. Management of the Disease

##### The Care Process: Difficulties in Community/Hospital Coordination and the Home Care Network

The healthcare professionals reported a lack of information transmission and a compartmentalisation between community and hospital care. At home, the network and the organisation of home care seemed to rely mainly on nurses who were very involved with their patients and tried to compensate for the lack of GPs. As a result, many situations resulted in “task shifting” for them, with the assessment and implementation of often complex medical issues. This resulted in very time-consuming work situations and a major risk of burnout for these professionals.

##### Medical and Technical Resources

A frequent observation was that at home or in hospital, there seemed to be no lack of technical or medicinal resources (notably access to opioids, although oxycodone is difficult to dispense). However, this finding is qualified by the lack of a suitable hospital service and the absence of possible recourse to a specialised team.

##### Diseases Treated at Advanced Stages, with Pain as the Main Symptom

Healthcare professionals described the frequency of late diagnoses, particularly of neoplastic diseases, leading to “dramatic” situations. This concerned the population from the Comoros but also the older generations of Mahorans, who are used to seeking care late.


*“N: it’s patients who come in the terminal stage and who are not from here and are not followed up here, but who come from the Comoros, and there they are already in an advanced state.”*
(interview S19)

The majority of patients mentioned pain as the most frequent and most distressing symptom during the course of the disease, sometimes with long delays in treatment at home due to the lack of medical follow-up. They could thus find themselves powerless and affected by the situations encountered.

#### 3.2.3. Beliefs and Culture in Mayotte

##### Religion and Culture Are Not an Obstacle to Care

Religion played a major role in a population that was almost exclusively Muslim. This role even became predominant when medical knowledge did not offer any hope of curing the disease. The notion of medical power was always supplanted by divine power. However, even if the religious dimension was omnipresent, all the health workers told us that it did not hinder the treatment. The patients agreed with the proposed care plans. Similarly, in the context of palliative care, the use of traditional medicine could be maintained but it generally did not interfere with care. Healthcare providers generally viewed it favourably because its use made sense for the patient. It was reserved either for certain specific types of care or for when it was recognised that “the hospital” could “do nothing more”.

##### A High Cultural Impact on the Understanding of the Disease and the Limitations of Interpretation

Doctors told us that there was a high degree of cultural variability in the understanding of disease, which had to be taken into account. For example, the concepts of chronic or genetic diseases were not part of the general population’s way of thinking. Additionally, the diagnosis of cancer was often perceived as a near-death situation.


*“N: as soon as they hear the word “cancer”, they already think of death. Because the diagnosis is made very late, maybe that’s why they have in mind that cancer equals death or serious illness equals death. We try to explain it to them, but […] they say “No, the doctor never told me that”, whereas often the family comes in and says: “Yes, yes, we know about it.”*
(interview S1)

The difference in language could also be a source of misunderstanding. Indeed, very few Mahorans spoke French properly, yet most of the doctors came from mainland France. Generally speaking, the healthcare professionals told us that patients did not necessarily dare to ask questions about their treatment, which was a sign of respect for the “knowing” doctor. They were thus often perceived as passive.


*“D: I find that they don’t ask for anything, it’s just culturally, people don’t complain […] It’s obvious that they don’t ask for anything, after that they have needs for almost everything, since there is almost nothing, in palliative care.”*
(interview S3)

##### Families, Privileged and Very Involved Interlocutors

Families had a special place, often at the first line of discussion with the doctors. Sometimes a family member acted as a referent and became the privileged interlocutor of the healthcare professionals. It was common to see families deciding to withhold medical information in order to maintain the patient’s hope and morale:

“N: *There are families […] who say ”It’s better you don’t tell the patient”. So most of the time, it’s the family that knows but not the patient himself.*”(interview S1)

The involvement of the families was more than an accompaniment in the care process, it was a participation in care, including heavy care, sometimes to the point of exhaustion.

“P: *I don’t have a sister, so I don’t have anyone to look after me, only my son does what he can, he’s a schoolboy and between school and looking after me it’s a bit of a struggle, I mean… F: I stay with my mother and soon I’m going to start preparing for the gendarme exam…*”(interview P4)

##### A Perception of Death That Is Far Removed from the “Classic” European Representations

The feedback from the survey was unanimous: death was not a subject to be discussed directly with Mahoran patients and their families. Some healthcare professionals mentioned a “cultural taboo” that had to be respected but which could make the palliative approach difficult to explain. For the patients and their families, the doctor had to seek a cure and it was not his role to announce the occurrence of death. In the terminal phase, the presence of members of the community, reciting prayers around the patient, signified the understanding of the relatives that death was near. Therefore, the concept of anticipatory discussion was rarely appropriate.

#### 3.2.4. The Health Professionals’ Proposals

The health professionals’ main proposal was the creation of a mobile structure dedicated to the care of patients in palliative situations, both in and out of hospital, with the following criteria:-Having a “medical referent” role. This role would require a high degree of availability both to patients and their families and private healthcare providers.-Whatever the social situation of the patients may be (affiliated or not to the social security).-Having a coordinating role in care to avoid any break in the care plan, to enable better interaction between the hospital and community healthcare providers (nurses, doctors, social workers, etc.). For example, it was proposed that a doctor from the palliative care structure should be present as soon as the incurable disease was announced.-A multidisciplinary team with a role in supporting the healthcare teams in their decision making by taking part in the definition of the care project, participation in staff discussions and ethical aspects.

The other proposals were as follows:-Suggestion to create a home hospitalisation service as a complement, which would also fulfil the condition of receiving patients not affiliated to the social welfare system, thus a public structure.-Creation of dedicated hospital palliative care beds in a department where the care team was specifically trained and equipped to accompany patients.-Providing enhanced training in several areas (palliative care, pain management, delivering bad news, interculturality and translation).

## 4. Discussion

### 4.1. The Problem of Legal Status and Access to Social Welfare

This aspect of our research was mentioned several times during the interviews. Undocumented people expressed a number of needs specific to them, linked to their administrative situation and their precariousness. This central element seemed fundamental to us because of its exceptional nature in a French department. It is consistent with the general demographic data provided in the literature, with a population made up of almost 50% foreigners, the majority of whom are Comorian, half of whom are in an irregular administrative situation [1,24]. It can be assumed, based on what the healthcare workers stated and without being able to assess the real numerical importance (due to the absence of exhaustive identification of these patients), that this population of patients is substantial in terms of being cared for by the health system. Although health reasons are not usually ranked as the main reason for immigration to Mayotte [10], the patients who were interviewed all stated that the disease, which appeared while they were living in their country of origin, was the only reason they came to Mayotte for treatment. In accordance with the data in the literature, our study confirms the experience of these patients with extremely precarious living conditions at all levels, experiencing a feeling of insecurity on a daily basis, the fear of expulsion and difficulties in accessing care [3,4,25].

In addition, in general terms in the published literature, difficulties in accessing social benefits are regularly mentioned, as in our study. Mayotte is described as “*the poorest French department, the most unequal and the most affected by unemployment*” [26].

### 4.2. Management and Care Process

#### 4.2.1. The Diseases

It is often at a very advanced stage of their disease that patients are treated. Delayed access to care is commonly accepted in the field. However, there is little recent data in the literature. Older data confirm the analysis: Florence et al. [27] identified the factors that hindered access to healthcare in a population of more than 2000 Mahorans representative of the general population in 2007; 70% of those interviewed said that they encountered obstacles to seeking care. For the locals, the main obstacle was waiting for a consultation (61%), followed by financial problems (29%). For the foreign population, it was primarily financial reasons (78%), followed by the fear of being arrested (41%). With regards to the situation of children in Mayotte, it is noted that there is a delay in accessing healthcare, due to the fact that minors do not have access to social welfare [26]. The non-governmental organisation Médecins du Monde notes that “*the specific health system in Mayotte doesn’t respect the International Convention on the Rights of the Child. Direct access to the children’s social welfare system would reduce the precariousness of immigrants children’s health*”.

In our study, pain was clearly the most frequent symptom. The literature is poor on this subject, although health professionals feel they have the means for “modern” healthcare, the qualitative survey reveals a lack of access to certain molecules (for example, oxycodone and trans-mucosal fentanyl), the lack of training in the assessment and management of pain, and the lack of medical follow-up at home or access to medical techniques such as Patient Controlled Analgesia (PCA) due to the absence of home care packages (HAD).

#### 4.2.2. Care at Home

Home care suffers from a serious lack of follow-up by GPs, which affects the care plan. Indeed, all recent data show that there is a major shortage of private GPs (only 24 doctors listed in 2019). In the 2016 Senate report [10], the supply of private health care is described as “non-existent”. Our study forcefully reveals this deficiency felt by patients, their families and the health professionals. The importance of the commitment of private nurses, which is one of the consequences, can be a source of professional burn out. The hospital in Mayotte provides medicines almost free of charge to meet the needs of those who do not have access to social welfare. However, the absence of a HAD limits the possibility of using hospital dispensed treatments at home (injectable paracetamol, midazolam, etc.) or techniques requiring specific monitoring (e.g., PCA).

#### 4.2.3. Hospital Care

Despite certain dysfunctions (e.g., waiting times or overall organisation) and staff shortages, the hospital is described as the “heart” of the health system, benefiting from “*a complete technical platform*” and it compensates for the lack of community healthcare facilities: “*Facing the shortcomings of the private sector, the Mayotte hospital is by far the first provider of care in the territory, and is forced to carry out much broader missions than those usually provided by hospital establishments*” [10]. Nevertheless, there seems to be a significant “*leakage rate*”, recognised but difficult to quantify, of Mahorans to other territories (La Reunion, another French overseas department, or mainland France) for treatment when they have the opportunity to do.

#### 4.2.4. The Importance of Traditions, Beliefs and Culture in Care

The specificities found in the study and confirmed in the literature [8] lead us to seriously consider cultural and religious values in Mayotte in the future provision of care: the notion of the patient belonging to his or her social group with an impact on care, notions of solidarity and community with the constant presence of family referents involved in care and decision making, coexistence of traditional medicine, relationship to the end of life and death [7,28]. The healthcare providers should also be better trained in these specificities, and the more frequent participation of trained interpreters seems obvious.

### 4.3. Public Feedback

At the end of the research work, we submitted our proposals for palliative care provision in Mayotte to stakeholders, decision makers and funders. This meeting allowed us to explain the conclusions of the research, to present the results of the work to the relevant stakeholders and to consolidate the support of the local decision makers for the project.

The main proposal was the creation of a mobile palliative care team with the following specificities: a strengthened workforce, due to the cultural and social realities of the island, by the involvement of interpreters, spiritual referents and social support skills. The proposed structure should be established in Mamoudzou, as part of the departmental hospital, a pillar of the health system in Mayotte. However, the idea of developing decentralised structures, run by the hospital in the reference centres, was accepted. The idea here was to train local referents, with dedicated time within the team. The team’s scope of action was to go far beyond the usual framework of a mobile palliative care team, which in France is essentially a consultancy and expertise team. This aspect would be central but would be complemented by front-line interventions, including monitoring of the patient, his or her family and the prescriptions made, as well as coordination. Therefore, the creation of a home hospitalisation structure should also be envisaged to support home care and the implementation of complex treatments. Finally, a global training plan with several components and more specific actions should be implemented quickly.

### 4.4. Limitations and Strengths of the Study

The researchers focused on enhancing the internal validity of the study by limiting interpretation bias through a process of triangulation: by multiplying the interviews with several sources (three distinct populations studied: patients, relatives and healthcare professionals), the reliability of the results was thus improved [29,30]. In addition, the use of different methods (individual interviews and focus groups) allowed for a greater diversity of results [31]. The large number of people interviewed and the verification of the results by means of feedback from a large number of health professionals were also intended to compensate for this fragility [32].

The study was conducted by two researchers not involved in the health system in Mayotte and practising in another French overseas department. This reinforced the independent nature of the work of conducting the interviews and analysis by limiting any influence on the results. The involvement of the referrer made it possible, beyond the logistics, to promote the creation of a climate of trust with the populations studied. He was independent, had no hierarchical link with the healthcare workers interviewed, and was not directly involved in the care of the patients and their relatives, which limited reporting bias.

A bias in the selection of patients led to the construction of a sample including a majority of patients in an illegal situation on the territory. Additionally, the cancer etiology is almost exclusive within the population studied. Several factors may be responsible for these phenomena: an over-representation of the immigrant population suffering from serious pathologies of cancer origin (health migrations), a departure of the Mahorans from the department to seek treatment, and situations that are particularly striking for the healthcare professionals. The survey of caregivers, but especially the healthcare professionals’ survey, because of its broad scope, made it possible to receive a wide variety of views. This is consistent with the aim of qualitative studies to produce sufficient data to answer the research question [18]. To reinforce the relevance of extending the results to the whole population of the island, a larger number of patient and caregiver interviews would have been interesting. It would also have been interesting to complement this with a quantitative study, for example, by questionnaire.

Language differences are also a source of interpretation bias. We tried to limit this bias through the participation of Mahoran interpreters trained in the study protocol, and through the participation of the referrer, who has experience in analysing disease representations in Mayotte.

It was rare to be able to meet patients alone, in their own environment, without a relative being present. However, we believe that this feature of our surveys had little impact on the verbatims of the interviewees. Furthermore, the interviews did not generate negative feedback and participants did not seek the psychological support offered to them.

At no time did we consider, with the exchanges between the members of the scientific committee, that the interview guides needed to be re-evaluated, although we had the possibility of doing so in our protocol.

## 5. Conclusions

This study achieved its main objective, which was to determine the needs of the population of Mayotte in the area of palliative and end-of-life care. The researchers were thus able to use the results to make a proposal for palliative care provision adapted to the context of Mayotte and its population. This was the secondary objective of the study. The methodological approach, published in 2021, was based on regular exchanges between the various stakeholders in the field and, through a process of co-construction, helped to encourage the adhesion of the healthcare practitioners and institutional decision makers. The study had an immediate concrete impact through the commitment of the management of the Mayotte Hospital Centre and the Regional Health Agency to create a mobile palliative care team at the end of 2019. The knowledge provided by the results of the study will enable this structure to be original and innovative, taking into account the specific needs of the Mayotte population. In addition, the results of the study could be particularly useful for government or NGO health programmes in the neighbouring islands of Madagascar and Comoros. Finally, the remote evaluation of the relevance of the proposals and the impact produced by their implementation is envisaged and will require the continuation of the research with the creation of relevant indicators. Replication of all or part of the study in a few years’ time could lead to a possible readjustment of the means implemented.

## Figures and Tables

**Figure 1 ijerph-19-03540-f001:**
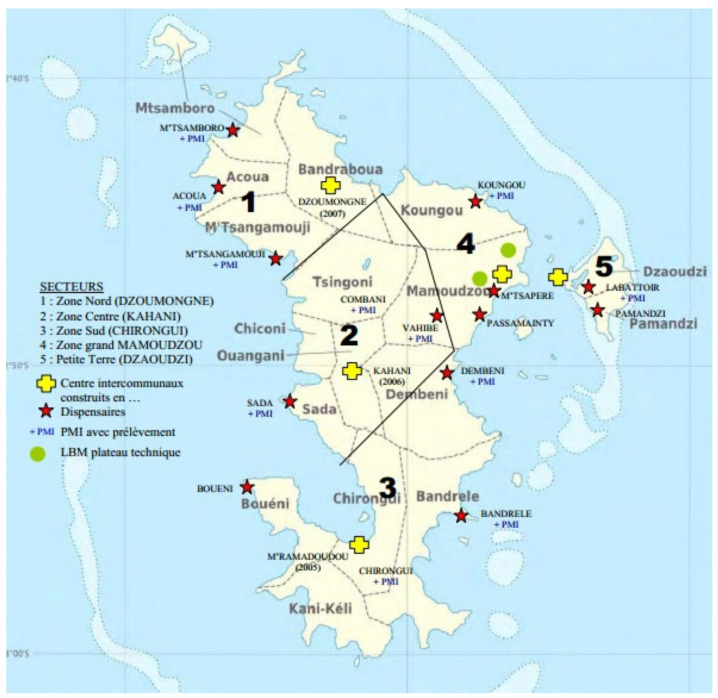
Health provision in Mayotte (source: Mayotte Regional Health Agency). Yellow crosses: reference hospitals built in (date), red stars: dispensaries, green dots: technical platforms, PMI: maternal and paediatric follow-up centre.

**Table 1 ijerph-19-03540-t001:** Description of the population.

Patients (9) for a Total of 13 Identified Patients			
Age (years)	Average: 42 Range: 21–78	Gender	Female: 6 Male: 3
Welfare affiliation	Yes: 2 No: 7	Diseases	Cancer: 8 Organ failure: 1
In total: 13 situations were identified and 9 interviews were conducted. The decision not to interview included three patients (a young child who could not make himself understood, a patient who could not be heard due to a tumor of the facial sphere and one patient refused to be recorded but agreed to be included so that his family could be interviewed). One patient had died after the first consent; his family and healthcare providers were interviewed.			
Families and other caregivers (15)			
Relation		Children: 8, Spouse: 1, Brother or sister: 2, Other: 4	
Nationality		French: 8, Comorian: 7	
Healthcare professionals (38)			
Interviews		Focus group: 7 (including 3 dyads), Individual: 7	
Professions		Nurses: 14, Doctors: 10, Nursing assistants: 9, Physiotherapists: 2, Psychologists: 2, Managers: 1	
Mode of practice		Hospital: 25, Community healthcare settings: 13	

**Table 2 ijerph-19-03540-t002:** Synthesis of the needs expressed.

**Patient and Family Survey**
The question of administrative status and non-affiliation Access to home care (population without social welfare) Ensure equal access to health care, facilitate approaches to social services Improve daily living conditions (material and financial) and stop living in hiding Management of the disease Improve pain management and access to medications at home Improve coordination between healthcare professionals Facilitate doctor’s visits to the home Beliefs and culture in Mayotte Improve the possibilities and quality of translations Help healthcare professionals to better respect the Mahoran culture and traditions (especially in the end of life) Maintain the coexistence of French and Mahoran culture
**Healthcare Professionals Survey**
The question of administrative status and non-affiliation Access to home care (especially medical follow-up by the community teams) Facilitate access to social assistance Access to appropriate care in acceptable conditions without interruption until the end of their illness Management of the disease Improve and develop coordination between healthcare professionals and in home care Facilitate medical follow-up by a doctor at home Support and advise professionals to better care for patients (nursing and medical expertise, psychological help and care relay) Beliefs and culture in Mayotte Assist healthcare professionals to better understand local culture and traditions Adapt the approach to care to the culture of the patient and the community The health professionals’ proposal Creation of a mobile palliative care team, a community intervention team and palliative care hospital beds Training programmes in several areas (pain management, interculturality and translation)

## Data Availability

The data sets used and analysed during the study are available from the first author (R.B.) upon written request and in accordance with ethical approval.

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
