# Peer review of "Palliative Care Needs Assessment in the Population Living in Mayotte: SPMAYOTTE, a Qualitative Study Conducted with 62 Patients, Caregivers and Healthcare Professionals"

_ijerph, 2022, doi:10.3390/ijerph19063540_

Round 1

Reviewer 1 Report

Thank you for a well-structured manuscript.

Please try to address the following observations with the aim of further enhancing your manuscript.

Introduction: There is no mention of NGO's or initiatives by international organizations - perhaps these have not happened in the field of palliative care, of are they completely absent in the field of healthcare on the island? The reason is that in other island communities, NGOs participation is seen as complementary to healthcare provision.

Discussion: The results are well discussed within the Mayotte context, however there is little comparison to healthcare provision on neighbouring islands e.g. Madagascar, where beliefs might be comparable - but healthcare provision different. Please enrich the discussion through such comparisons where possible .

Reviewer 2 Report

This manuscript is an original article that aimed to determine the palliative care needs of the Mayotte population and make a proposal for setting up an appropriate service using a qualitative study conducted with 62 patients, caregivers and healthcare professionals. The authors extracted several problems from interviews and discussed them.

This study was conducted well, and the methods are appropriate. The results will be of interest to clinicians in the field.

However, the following minor issues require clarification:

Minor

  1. Study population, especially patients and caregivers, seems too small to comprehensively assess palliative care needs in Mayotte. I recommend that the authors add this in the limitation.
  2. Please provide origins of cancer.
  3. It’s difficult to understand Table 2. Please explain it in more detail.
